# UPF1 Inhibits Hepatocellular Carcinoma Growth through DUSP1/p53 Signal Pathway

**DOI:** 10.3390/biomedicines10040793

**Published:** 2022-03-29

**Authors:** Suman Lee, Yukyung Hwang, Tae Hun Kim, Jaemin Jeong, Dongho Choi, Jungwook Hwang

**Affiliations:** 1Graduate School for Biomedical Science & Engineering, Hanyang University, Seoul 04763, Korea; sumanlee@hanyang.ac.kr (S.L.); dbrud1151@naver.com (Y.H.); 2Department of Surgery, Hanyang University College of Medicine, Seoul 04763, Korea; xo132435@hanyang.ac.kr (T.H.K.); jmj1103@hanyang.ac.kr (J.J.); crane87@hanyang.ac.kr (D.C.)

**Keywords:** UPF1, hepatocellular carcinoma, posttranscriptional regulation, DUSP1

## Abstract

Human hepatocellular carcinoma (HCC) has a high mortality rate because of the dearth of effective treatments. Multiple studies have shown that overexpression of UPF1, a key nonsense-mediated mRNA decay (NMD) factor, reduces HCC growth through various cell signaling pathways. However, the mechanism by which UPF1 expression retards HCC proliferation through the regulation of RNA stability remains unclear. By employing various UPF1 variants and transcriptome analysis, we revealed that overexpression of UPF1 variants, not UPF1-mediated NMD, reduces HCC tumorigenesis. Additionally, UPF1 variant overexpression reduced tumorigenesis in xenografted mice. Transcriptome analysis indicated that the level of dual specificity phosphatase 1 (DUSP1) was increased by UPF1 variants via posttranscriptional regulation. The UPF1 overexpression-mediated increase of DUSP1 activated tumor suppressor signaling, ultimately inhibiting cell growth. In this study, we highlighted the function of UPF1 as a tumor suppressor in HCC growth.

## 1. Introduction

UPF1 plays a critical role in nonsense-mediated mRNA decay (NMD) by removing premature termination codon (PTC)-containing transcripts. PTCs result from DNA rearrangement, DNA mutation, frameshifts induced by alternative splicing, and untranslated open reading frames (uORFs), as well as introns in the 3′ untranslated region (3′UTR), in which a normal translation termination codon functions as a PTC. After recognition of a PTC by a translating ribosome, UPF1 interacts with various NMD factors, including eRF3, UPF2, SMG1, SMG5 and SMG6, leading to mRNA decay [1,2]. Because nonsense mutations are observed in cancer-related oncogenes and tumor suppressor genes, producing C-terminal truncated proteins when NMD fails, cancer may arise from nonfunctional NMD. Indeed, nonsense mutations in adenomatous polyposis coli (APC), which suppresses Wnt signaling as a tumor suppressor, increase the risk of sporadic and hereditary colorectal cancer [3]. Furthermore, multiple transcripts, including TP53, ALB, AXIN1 and BAP1, which play a key role in hepatocellular carcinoma (HCC), exhibit nonsense mutations, suggesting the importance of NMD function [4].

HCC is a leading cause of cancer-related death worldwide [5]. The main reasons for the high mortality of HCC are that surgery is restricted to a small number of patients and effective drug treatment is lacking. Although studies have specifically addressed the molecular mechanism of HCC and recent advances have been made in medical treatment, the varied effectors of carcinogenesis, such as viruses, sources of metastasis and genetic mutations, make it difficult to determine optimal treatment. Recently, several studies have pointed out that UPF1 is a potential tumor suppressor in HCC. UPF1 directly interacts with snoRNA host gene 6 (SNHG6), resulting in the promotion of ZEB1 expression and upregulation of TGFβ1/Smad signaling [6]. Interestingly, UPF1 expression was significantly downregulated in HCC tissues derived from patients, and high expression of UPF1 increased the survival rate and upregulation of UPF1-conferred sensitivity to sorafenib in HCC [7,8]. However, whether UPF1 affects HCC growth through NMD remains unclear.

Among various tumor suppressors, dual specificity phosphatase 1 (DUSP1) inhibits ERK signaling and inhibits cell cycle progression in HCC cells [9]. Significantly higher expression of DUSP1 was observed more frequently in patients with higher overall survival than lower overall survival. Furthermore, overexpression of DUSP1 increased phos-p53, p21 and p27, and these genes are well-known antitumor genes in cancer [10,11,12,13]. Taking these reports into account, DUSP1 may regulate HCC cell proliferation.

To clarify the function of UPF1 in HCC proliferation as a key NMD factor, we first evaluated the effects of UPF1 overexpression on NMD and Huh7 cell growth. Overexpression of UPF1 did not change levels of exogenous and endogenous NMD targets but did retard Huh7 cell growth. Moreover, overexpression of UPF1 deletional variants with loss of the key domains for NMD also reduced Huh7 tumorigenesis. All in vitro results were confirmed in a xenograft model, indicating that UPF1 variant-expressing tumor tissues exhibited loss of tumorigenesis. Transcriptome analysis of cells with UPF1 overexpression or depletion identified the potential UPF1-targeted transcript DUSP1, which is posttranscriptionally regulated by UPF1. Our studies revealed that the growth of liver cancer can be slowed by UPF1 variant overexpression, but not through UPF1-mediated NMD.

## 2. Materials and Methods

### 2.1. Cell Culture, Transfection and Infection

Huh7, PLC/PRF/5, SNU-354, SK-HEP-1, HepG2 (all cells were purchased from Korean Cell line bank) and 293GPG [14,15] cells were maintained in Dulbecco’s modified Eagle’s medium (DMEM) supplemented with 10% fetal bovine serum and 1% penicillin/streptomycin. To silence or overexpress the target gene, Huh7 cells were seeded one day before DNA or siRNA transfection with Lipofectamine 3000 (Invitrogen; Waltham, MA, USA). For retrovirus production, EGFP-IRES-C5W2 plasmids containing UPF1 variants were transfected into 293GPG cells for 7 days. Cell medium supernatant was collected, and viruses were concentrated using PEG buffer and polybrene. The specific siRNA (IDT Technologies; Coralville, IA, USA) sequences are listed in Appendix A. To test posttranscriptional regulation, transcription was inhibited with DRB. Huh7 cells were treated with 100 nM of sorafenib (European Pharmacopoeia Reference Standard; Strasbourg, France) to induce apoptosis as a positive control [16].

### 2.2. Plasmid Construction

To construct the EGFP-MYC-UPF1 WT plasmid, the EGFP-IRES-C5W2 vector and the PCR product from pCMV-MYC-UPF1 amplified with the primers UPF1-BamHI-F (1-1118) and UPF1-BamHI-R (1-1118) were digested with BamHI and ligated [17]. Similarly, the UPF CH and UPF1 HD fragments were amplified with primer pairs UPF1-XhoI-F (1-244)/UPF1-NotI-R (1-244) and UPF1-XhoI-F (295-914)/UPF1-NotI-R (295-914), respectively, from the corresponding DNA templates pCMV-MYC-UPF1 (1-244) and pCMV-MYC-UPF1 (295-914) [17]. Vectors and PCR products were digested with XhoI and NotI and then ligated. The cloning primer sequences are listed in Appendix A.

### 2.3. Western Blotting (WB) Analysis

To detect specific proteins, proteins in total cell lysates were eluted with SDS and β-mercaptoethanol. Then, the proteins were separated on gels containing various percentages of polyacrylamide and transferred to nitrocellulose membranes. Blotting was performed with antibodies specific for the following proteins: MYC (ThermoFisher; Waltham, MA, USA), UPF1 (Cell Signaling; Danvers, MA, USA), calnexin (Santa Cruz; Dallas, TX, USA), β-actin (Sigma; St. Louis, MO, USA), GFP (Santa Cruz; Dallas, TX, USA), DUSP1 (Cell Signaling; Danvers, MA, USA), p53 (Santa Cruz; Dallas, TX, USA), phospho-p53 (Ser15) (Cell Signaling; Danvers, MA, USA), p27 (Cell Signaling; Danvers, MA, USA), caspase-3 (Cell Signaling; Danvers, MA, USA), cleaved caspase-3 (Cell Signaling; Danvers, MA, USA), caspase-9 (Cell Signaling; Danvers, MA, USA), cleaved caspase-9 (Cell Signaling; Danvers, MA, USA), PARP (Cell Signaling; Danvers, MA, USA) and cleaved PARP (Cell Signaling; Danvers, MA, USA).

### 2.4. RT-qPCR

Total RNA was extracted using TRIzol (Invitrogen; Waltham, MA, USA). To remove exogenous and endogenous DNA, extracted RNA was treated with RQ DNase I (Promega; Madison, WI, USA). cDNA was synthesized with RTase (ThermoFisher; Waltham, MA, USA) using random hexamer primers (Macrogen, Seoul, Korea). RT-qPCR was performed using the primers listed in Appendix A.

### 2.5. Calculation of NMD Efficiency

To determine NMD efficiency, cells were transfected with NMD test plasmids [pmCMV-Gl (β-globin) or pmCMV-GPx1 (glutathione peroxidase 1), either PTC-free (Norm) or PTC-containing (Ter)] and a reference plasmid [phCMV-MUP (mouse major urinary protein)]. The fold change in NMD was calculated as the ratio of test NMD (for the PTC-containing gene) to control NMD (for the PTC-free gene), where NMD is the relative abundance of Ter mRNA divided by the relative abundance of Norm mRNA. The abundance of MUP mRNA served as the reference.

### 2.6. Migration and Invasion Assays

To evaluate cell migration, layers of Huh7 cells infected with retroviral vectors expressing UPF1 variants were scratched with a sharp tip, and migration was evaluated with AxioVision Rel 4.8 software. Similarly, infected cells were resuspended in serum-free medium and seeded in an invasion chamber insert with a Matrigel-coated membrane (Corning; Corning, NY, USA). After 24 h of incubation, cells that had penetrated the membrane were visualized by hematoxylin and eosin staining.

### 2.7. Immunofluorescence and Immunohistochemical Staining

Cells seeded on round microscope coverslips were fixed with 3.7% formaldehyde in PBS for 20 min at room temperature and permeabilized with PBS containing 0.2% saponin three times for 5 min each. Then, the cells were blocked with PBS containing 0.2% saponin and 3% BSA and incubated for 1 h at room temperature. The fixed and blocked cells were incubated with primary antibodies in PBS containing 0.2% saponin and 3% BSA overnight at 4 °C. After three washes in PBS containing 0.2% saponin, the cells were incubated with secondary antibodies (Alexa 488 or Alexa546, Life Technologies; Carlsbad, CA, USA) diluted 1:1000. The cells were washed with PBS containing 0.2% saponin 3 times for 5 min each and with PBS one time for 5 min. After washing, the coverslips were mounted with Vectashield mounting medium with DAPI (Vector Laboratories, Inc. #H-1200, Burlingame, CA, USA). Each protein was visualized with the corresponding primary antibody and fluorescent secondary antibody. Similarly, in vivo tumors were evaluated by immunohistochemistry. Paraffin sections were deparaffinized, rehydrated, subjected to antigen retrieval, and blocked with peroxidase. Stained tumor sections were visualized by incubation in 3,3′-diaminobenzidine (DAB) buffer and were then counterstained, dehydrated, cleared, and mounted. The sections were then incubated with antibodies specific for the following proteins: PCNA (Santa Cruz; Dallas, TX, USA), Ki67 (Cell Signaling; Danvers, MA, USA), DUSP1 (Santa Cruz; Dallas, TX, USA), phospho-p53 (Ser15) (Cell Signaling; Danvers, MA, USA) and p27 (Cell Signaling; Danvers, MA, USA).

### 2.8. Flow Cytometry

To evaluate the cell cycle in UPF1 variant-infected Huh7 cells, a 5-bromo-2′-deoxyuridine flow kit (BD Pharmingen; San Diego, CA, USA) was employed. Three days after transduction, cells were incubated with 10 μM BrdU for 1 h prior to fixation. Then, 7-aminoactinomycin D (7-AAD) was added. Cell cycle phases were determined using flow cytometry (FACSCanto, BD Pharmingen; San Diego, CA, USA).

### 2.9. TUNEL Assay

For the TUNEL assay, we used the In Situ Apoptosis Detection Kit (TakaRa; Kusatsu, Japan) according to the manufacturer’s instructions. Briefly, cells seeded on round microscope coverslips were fixed in 4% paraformaldehyde for 15 min at room temperature, permeabilized by permeabilization buffer for 5 min on ice and incubated with TdT enzyme in labeling buffer for 90 min at 37 °C. After extensive washing with PBS, the coverslips were mounted with Vectashield mounting medium with DAPI (Vector Laboratories, Inc. #H-1200, Burlingame, CA, USA).

### 2.10. Xenograft Model

Eight female BALB/c nude mice (7 weeks old) were purchased from ORIENT BIO Inc. (Seongnam, Korea). Huh7 cells (1 × 10^6^ cells/spot) infected with retroviral vectors expressing each EGFP-MYC-UPF1 variant were injected subcutaneously into the flank of each BALB/c nude mouse. Eight weeks post-injection, mice were sacrificed, and tumors were harvested for IHC staining and weight measurement.

### 2.11. Statistical Analysis

Unpaired Student’s *t*-test was used to calculate *p*-values in RT-qPCR analyses. Differences with *p* < 0.05, 0.01, or 0.001, as indicated in the figure legends, were considered statistically significant. Mean and s.e.m. values were calculated from independent experiments. The Kolmogorov–Smirnov (K-S) test was performed for all cumulative distribution function (CDF) analyses.

### 2.12. RNA-Seq and Data Analysis

The total RNA concentration was calculated with Quant-IT RiboGreen (Invitrogen, #R11490; Waltham, MA, USA). A library was independently prepared with 1 µg of total RNA per sample with an Illumina TruSeq Stranded mRNA Sample Prep Kit (Illumina, Inc., San Diego, CA, USA, #RS-122-2101). The libraries were quantified using KAPA Library Quantification Kits for Illumina Sequencing platforms according to the qPCR Quantification Protocol Guide (KAPA BIOSYSTEMS, #KK4854; Wilmington, MA, USA) and qualified using a TapeStation D1000 ScreenTape chip (Agilent Technologies, #5067-5582; Santa Clara, CA, USA). Indexed libraries were then subjected to paired-end (2 × 100 bp) sequencing on the Illumina NovaSeq platform (Illumina, Inc., San Diego, CA, USA) by Macrogen Incorporated.

Before analysis, we preprocessed the raw sequencing reads to remove low-quality reads and adapter sequences and aligned the processed reads to the *Homo sapiens* genome (hg19) using HISAT v2.1.0 (Baltimore, MD, USA) [18]. The reference genome sequence of *H. sapiens* (hg19) and annotation data were downloaded from the NCBI database. Then, the assembly of known transcripts, novel transcripts, and alternative splicing transcripts was performed with StringTie v1.3.4d (Baltimore, MD, USA) [19,20]. Based on these results, the abundance of transcripts and genes was calculated as the read counts or Fragments Per Kilobase of exon per Million fragments mapped (FPKM) for each sample. The expression profiles were used to perform additional analyses, such as analysis of differentially expressed genes (DEGs). In groups with different treatment conditions, differentially expressed genes or transcripts can be filtered through statistical hypothesis testing. Gene functional classification and Gene Ontology (GO) analysis were performed using DAVID Bioinformatics Resources version 6.8 (Frederick, MD, USA) [21,22].

## 3. Results

### 3.1. UPF1 Variants Inhibit Huh7 Cell Growth

Multiple studies have shown that overexpression of UPF1 reduces the growth of HCC cells [7,23,24]. Because UPF1 is involved in NMD mainly as a posttranscriptional regulator, we postulated that UPF1 retards HCC growth by regulating NMD target expression. To determine the effects of UPF1 on HCC growth, we first infected Huh7 cells with retrovirus expressing wild-type (WT) UPF1 and quantified the cell number for 96 h (Figure 1A). Consistent with previous reports, overexpression of UPF1 delayed Huh7 cell growth. Because UPF1 plays an important role in NMD, it is conceivable that UPF1 regulates Huh7 cell growth through NMD. To confirm this hypothesis, first, we overexpressed WT UPF1 or UPF1 deletion variants, because the UPF1 domain itself cannot support the NMD mechanism (Figure 1B). We designed UPF1 deletion variants containing only the cysteine/histidine-rich domain (CH) or the helicase domain (HD), both of which are essential in UPF1 for the elimination of NMD targets. The CH interacts with NMD and Staufen-mediated mRNA decay (SMD) factors, such as UPF2, eRF3 and Staufen, and the HD has a helicase function to unwind double-stranded RNA [25,26]. To assess inhibition of Huh7 cell growth, we overexpressed Huh7 cells with UPF1 variants via retrovirus infection. Similar to overexpression of UPF1 WT, overexpression of either UPF1 variant (CH or HD) in Huh7 cells impaired cell growth (Figure 1C). These observations were reproduced in other HCC cell lines including PLC/PRF/5 and SNU-354, but not in HepG2 (hepatoblastoma) or SK-Hep-1 (cancer cells from the intrahepatic bile duct) (Appendix A). Interestingly, overexpression of UPF1 variants did not affect the level of well-known NMD targets (Figure 1D). These observations were confirmed by measuring the abundance of exogenous NMD reporters expressing Gl- or GPx1-containing PTC, indicating that overexpression of UPF1 variants did not have effects on NMD efficiency (Figure 1E). In addition, transcriptome analysis indicated that the expression of endogenous NMD targets increased in response to UPF1 depletion in Huh7 cells, but UPF1 overexpression did not show significant changes in endogenous NMD target expression (Figure 1F). Collectively, these results indicate that the regulation of Huh7 cell growth by UPF1 is independent of its NMD function and may be controlled by another mechanism.

### 3.2. Overexpression of UPF1 Variants Is Responsible for the Malignant Phenotypes of Huh7 Cells

Given the influence of UPF1 variants on Huh7 cell growth, we next sought to determine the effects of UPF1 variants on malignant phenotypes, including migration and metastatic potential, in Huh7 cells. Similar to the method used for the cell growth assay, Huh7 cells were infected with retrovirus expressing each UPF1 variant. Consistent with the cell growth assay results, overexpression of each UPF1 variant efficiently reduced cell migration (Figure 2A). However, on invasion assay, the UPF1 CH variant did not affect invasion ability, unlike UPF1 WT and the UPF1 HD variant, suggesting that overexpression of CH may only affect the growth and migration of cells (Figure 2B). Next, we investigated the expression of the cell proliferation marker proteins Ki-67 and PCNA in Huh7 cells infected with the UPF1 variants (Figure 2C,D). Immunofluorescence analysis indicated that overexpression of UPF1 variants generally reduced the expression of these cell proliferation markers; however, the UPF1 CH variant did not reduce PCNA expression. These observations led us to investigate the influence of UPF1 variant expression on the cell cycle. To investigate changes in cell cycle progression, we conducted fluorescence-activated cell sorting (FACS) analysis, which demonstrated that expression of UPF1 WT and UPF1 HD slightly increased the percentage of cells in the G1 phase compared with that in control and UPF1 CH-expressing cells (Figure 2E). Then, we examined UPF1 variants effects on apoptosis which may also result in growth retardation. The overexpression of UPF1 variants did not upregulate apoptosis markers including cleaved caspase-3, -9 and PARP, but sorafenib, as a positive control, did increase the levels of apoptosis markers (Figure 2F). Furthermore, the TUNEL assay demonstrated that DNA fragmentation did not occur during UPF1 variant expression, suggesting that overexpression of UPF1 variants did not cause apoptosis. Together, these results suggest that the expression of UPF1 variants induces cell cycle arrest, thereby, suppressing the oncogenic function of Huh7 cells.

### 3.3. Overexpression of UPF1 Variants Reduces Huh7 Tumor Growth In Vivo

To reproduce the effects of UPF1 variants on Huh7 cell tumorigenesis in vivo, Huh7 cells infected with retroviral vectors expressing the UPF1 variants were injected into nude mice and tumor volumes were monitored weekly for 8 weeks. The xenograft assay data showed that expression of either UPF1 WT or UPF1 HD efficiently inhibited tumor growth (Figure 3A–D), but UPF1 CH slightly decreased tumor volume and had no effect on tumor weight. Furthermore, overexpression of UPF1 variants dramatically reduced Ki-67 and PCNA expression in tumors (Figure 3E). In contrast to in vitro data, in vivo data showed that all UPF1 variants inhibited the expression of PCNA and Ki-67. The discrepancy between PCNA expression in vitro and in vivo may be explained by the involvement of other cell growth factors in tumor growth. The simple interpretation of the results is that UPF1 variants repress Huh7 tumorigenesis in vitro and in vivo.

### 3.4. UPF1 Expression Regulates the Level of DUSP1

We observed that overexpression of UPF1 variants efficiently abrogated tumorigenesis in Huh7 cells in vitro and in vivo but not through the NMD pathway. Because the main function of UPF1 in cells is as a regulator of posttranscription, we analyzed the transcriptome in Huh7 cells with UPF1 overexpression and UPF1 depletion. As expected, depletion of UPF1 directly or indirectly regulated the stability of many transcripts (*n* = 2482, *p* ≤ 0.05). In contrast, UPF1 overexpression in Huh7 cells did not change the abundance of many transcripts (*n* = 91, *p* ≤ 0.05) (Figure 4A). Interestingly, depletion of UPF1 upregulated some (*n* = 47) of the top 250 NMD targets, but overexpression of UPF1 had only minor effects on target stability (*n* = 1) among the top 250 targets (Figure 4A) [27]. These observations were consistent with the results of the analysis described in Figure 1F. Then, we selected the common transcripts that were upregulated and downregulated by depletion and overexpression of UPF1, respectively (Figure 4B). Among the selected transcripts, DUSP1 is a well-known tumor suppressor in HCC [9,10]. The results from GEPIA database analysis revealed that DUSP1 mRNA expression was significantly higher in normal liver tissues than in HCC tissues (Figure 4C). To examine whether UPF1 regulates the abundance of DUSP1 mRNA via posttranscriptional regulation, we measured DUSP1 mRNA stability under transcription inhibition conditions with the RNA polymerase II inhibitor 5,6-dichloro-1-beta-D-ribofuranosylbenzimidazole (DRB). The results of DRB treatment demonstrated that overexpression of UPF1 increased the half-life of DUSP1 mRNA, suggesting that UPF1 posttranscriptionally regulates DUSP1 mRNA stability (Figure 4D). These results were verified by overexpression of UPF1 variants in Huh7 cells by RT-qPCR, WB and immunofluorescence analysis indicating that UPF1 variant overexpression enhanced DUSP1 mRNA and protein expression (Figure 4E–G). Consistent with previous reports [10], DUSP1 upregulation by UPF1 variants increased p53 phosphorylation at S15 and slightly upregulated the expression of p27 by WB and immunofluorescence analysis (Figure 4F,G). Moreover, DUPSP1 expression was upregulated by overexpression of UPF1 variants in other HCC (PLC/PRF/5 and SNU-354) (Appendix A). Intriguingly, Huh7 and PLC/PRF/5 contained a mutation in p53, but SNU-354 had wild-type p53, suggesting that upregulated levels of DUSP1 and p27 were independent of p53 mutation. Thus, these results suggest that DUSP1 is regulated by UPF1 variants and that upregulated DUSP1 regulates the phosphorylation status of well-known tumor suppressor p53, leading to HCC growth retardation. Altogether, our findings imply that Huh7 cell growth is inhibited via the UPF1/DUSP1/p53 signaling pathway (Figure 4H).

## 4. Discussion

Liver cancer has a high mortality rate worldwide compared to other malignant tumors. The prevailing therapy for liver cancer is treatment with tyrosine kinase inhibitors (TKIs). However, TKI treatment cannot significantly eliminate HCC tumors with drug resistance [28]. Thus, a new approach for liver cancer treatment is urgently needed. To discover new treatments for liver cancer, we addressed the function of UPF1 in HCC. First, we demonstrated that overexpression of UPF1 inhibited the tumorigenesis of HCC cell lines including Huh7, PLC/PRF/5 and SNU-354 cells (Figure 1, Figure 2 and Appendix A). Because UPF1 has major functions in NMD, it is plausible that overexpression of UPF1 may change levels of endogenous NMD targets, ultimately leading to retardation of cell growth. However, we discovered that overexpression of UPF1 and its variants did not regulate levels of exogenous and endogenous NMD targets. Furthermore, overexpression of UPF1 variants changed the motility, invasion ability and cell cycle of Huh7 cells (Figure 2). All these results were reproduced in vivo. UPF1 variant-expressing Huh7 cells xenografted into mice grew slowly, leading to the formation of smaller tumors (Figure 3). Interestingly, overexpression of the UPF1 CH variant efficiently reduced Huh7 cell growth and migration to levels comparable to those of cells overexpressing UPF1 WT or UPF1 HD; however, overexpression of UPF1 CH had less noticeable effects on cell invasion and cell cycle progression (Figure 1 and Figure 2). These observations were consistent with the in vivo results (Figure 3). Overexpression of UPF1 WT and UPF1 HD in Huh7 cells significantly reduced the volume and weight of the corresponding xenograft tumors, but overexpression of UPF1 CH did not exert these effects. In summary, UPF1 CH efficiently reduced HCC cell growth, cell migration and tumor volume in vivo but had no effects on HCC cell invasion, cell cycle progression, tumor size, or tumor weight in vivo. One possible explanation could be that the UPF1 CH domain does not interact with other proteins, such as UPF1 binding partners. In this study, however, we did not address the detailed mechanism by which UPF1 variants suppress HCC tumorigenesis.

The results of RNA sequencing (RNA-seq) analysis were intriguing. Depletion of UPF1 in Huh7 cells upregulated many endogenous NMD targets (Figure 4). In contrast, overexpression of UPF1 did not exhibit any regulatory activity on NMD target levels, suggesting that the regulation of HCC growth by UPF1 overexpression is not due to altered levels of NMD targets. The abundance of DUSP1, which has antitumor effects in HCC, was posttranscriptionally regulated depending on the UPF1 expression level. Indeed, UPF1-overexpressing variants showed increased levels of DUSP1 and its downstream tumor suppressor proteins, including p53 phosphorylated at S15 and p27. In summary, we show the possible mechanism by which UPF1 variants regulate HCC tumorigenesis in Figure 4H. Expression of UPF1 and its variants upregulates DUSP1 expression via an increase in DUSP1 mRNA stability, which in turn increases levels of the indicated tumor suppressor proteins. Recently, gene expression for cancer therapy has been proposed through viral expression, such as adenovirus-associated virus (AAV), which is tissue specific [29,30]. The fact that UPF1 expression retarded HCC growth is beneficial in gene therapy for hepatocellular carcinoma. Because of the large molecule size of UPF1 (approximately 150 kDa), however, it is difficult to transmit full-length UPF1 using AAV. In this study, UPF1 deletion proteins had comparable functions in hepatocellular carcinoma inhibition to full-length UPF1, which would allow UPF1 variants to be employed for therapeutic approaches in hepatocellular carcinoma.

## Figures and Tables

**Figure 1 biomedicines-10-00793-f001:**
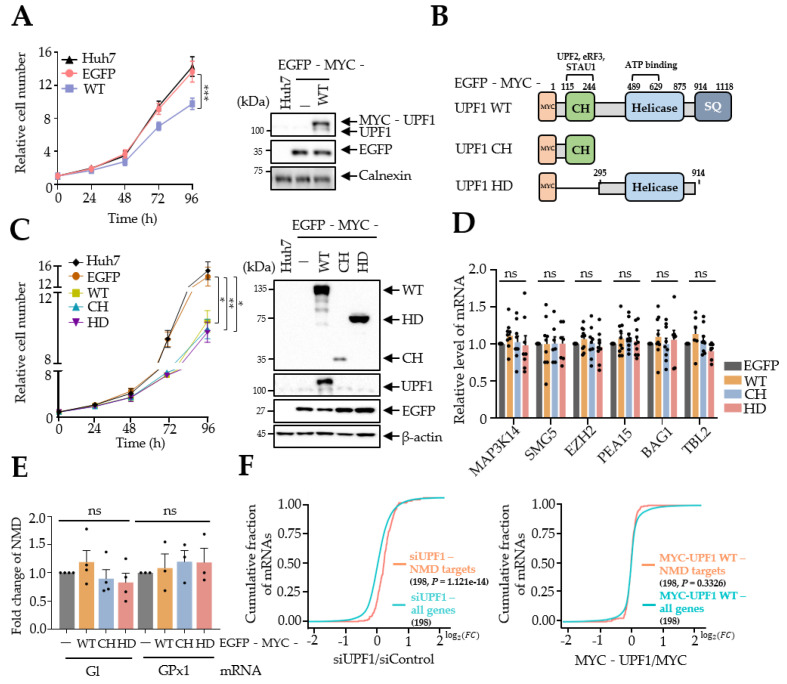
UPF1 deletion variants repressed Huh7 cell growth. (**A**) Huh7 cells that were infected with retrovirus expressing EGFP-MYC-UPF1 or EGFP as the negative control. Cell growth was measured by counting the cells over a period of 96 h. WB analysis was performed to evaluate the indicated proteins. (**B**) Schematic representation of UPF1 deletion variants. CH: cysteine/histidine rich region, HD: helicase domain, SQ: serine/glutamine region, MYC: myc tag. (**C**) Same as (**A**) except that Huh7 cells were infected with retrovirus expressing the indicated UPF1 deletion variants. Cell growth assays and WB analysis were performed. (**D**,**E**) NMD efficiency of endogenous NMD targets (**D**) and NMD reporters (**E**) was measured in UPF1 variant-expressing Huh7 cells by RT-qPCR. mRNA levels were normalized to those of GAPDH mRNA in (**D**) and MUP mRNA in (**E**). (**F**) Total RNA-seq was performed using UPF1-depleted or UPF1-overexpressing Huh7 cells. Cumulative log2 fold changes in expression in UPF1-depleted and UPF1-overexpressing cells compared with vehicle-transfected cells are shown as CDF plots using endogenous NMD targets. Indicated *p* values were estimated using the K-S test. The numbers in parentheses refer to endogenous NMD targets. * *p* ≤ 0.05; ** *p* ≤ 0.01; *** *p* ≤ 0.001; ns, not significant.

**Figure 2 biomedicines-10-00793-f002:**
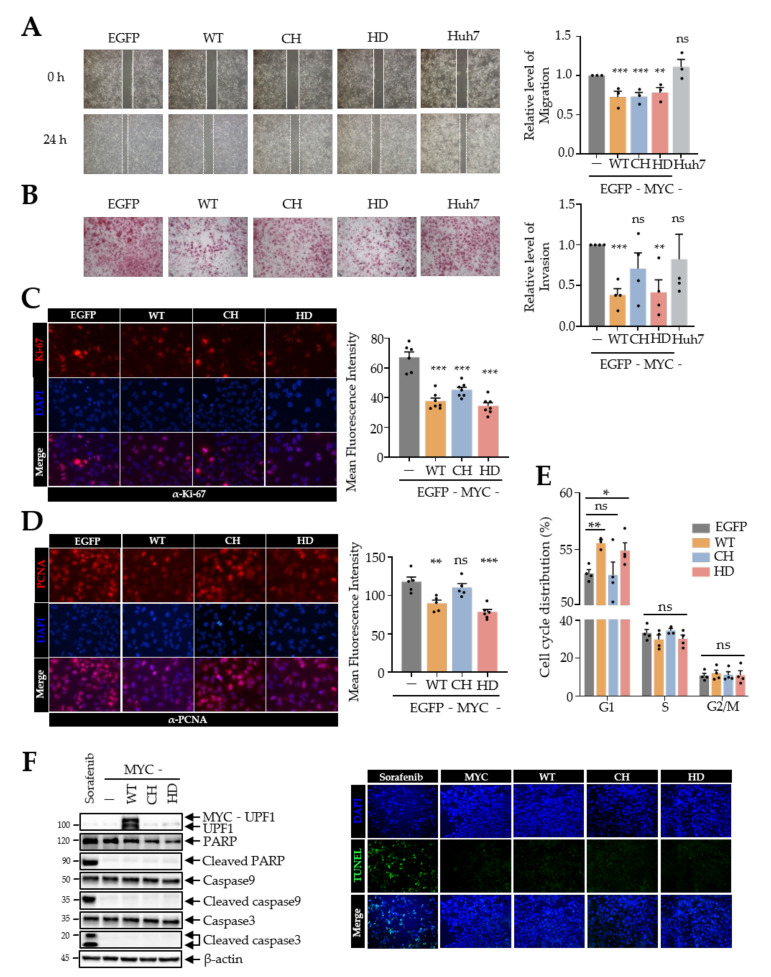
UPF1 variants reduced the tumorigenesis of Huh7 cells. Cell migration assay (100× magnification) (**A**), cell invasion assay (100× magnification) (**B**) and immunofluorescence analysis (**C**,**D**) were performed using Huh7 cells infected with retrovirus expressing UPF1 variants. Immunofluorescence microcopy showed Ki-67 (red) (**C**), PCNA (red) (**D**) and nuclei (blue) (400× magnification). Relative levels of migration, invasion, or expression in the assays were quantified. (**E**) The cell cycle distribution in UPF1 variant-expressing Huh7 cells was evaluated by flow cytometry. (**F**) Apoptosis analyses were determined by WB (left panel) and TUNEL assay (right panel) using UPF1 variant overexpressing Huh7 cells (200× magnification). * *p* ≤ 0.05; ** *p* ≤ 0.01; *** *p* ≤ 0.001; ns, not significant.

**Figure 3 biomedicines-10-00793-f003:**
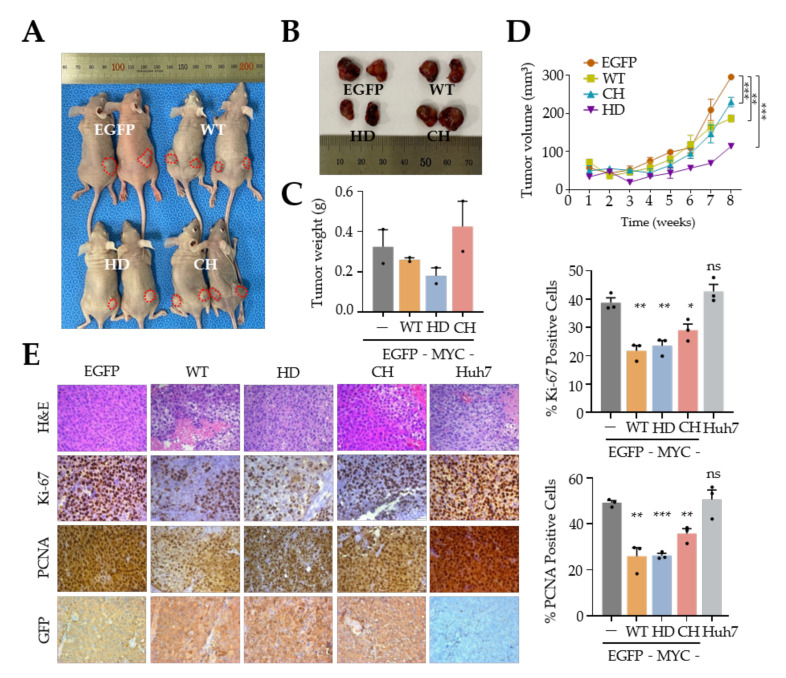
Overexpression of UPF1 WT and UPF1 HD attenuated tumor growth in vivo. (**A**) Representative images of tumors formed in nude mice grafted subcutaneously with Huh7 cells expressing UPF1 variants. (**B**) Representative tumors from grafted mice 8 weeks after cell implantation. (**C**) Tumors were weighed after removal (*n* = 2). (**D**) Tumor volumes were monitored weekly for 8 weeks. (**E**) Immunohistochemical staining for Ki-67 and PCNA in xenografted tumors in nude mice was performed (100× magnification). The relative numbers of Ki67- and PCNA-expressing cells were determined. * *p* ≤ 0.05; ** *p* ≤ 0.01; *** *p* ≤ 0.001; ns, not significant.

**Figure 4 biomedicines-10-00793-f004:**
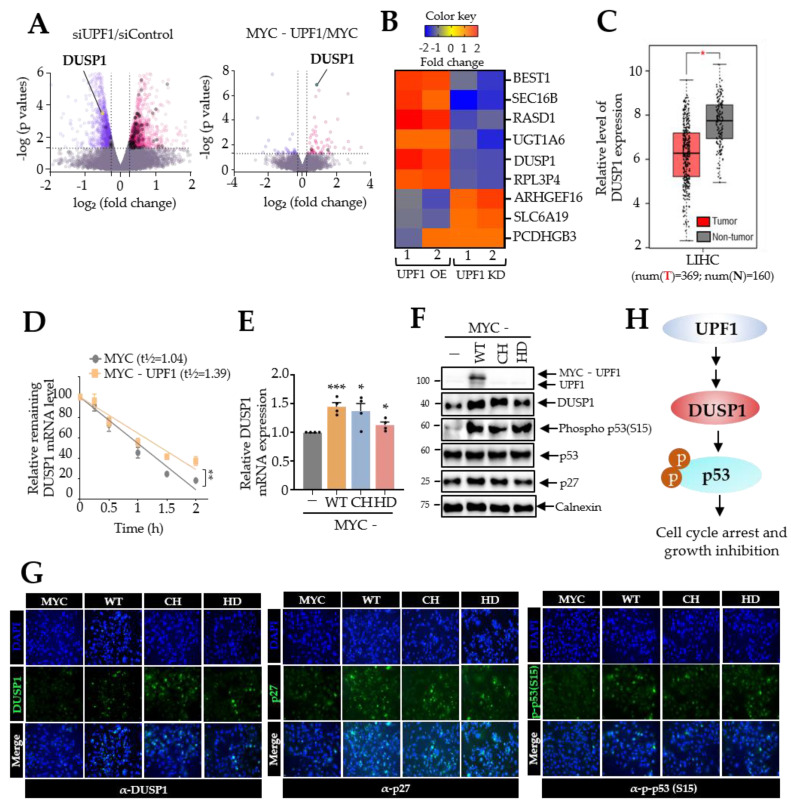
UPF1 regulates DUSP1 at the posttranscriptional level. (**A**) Volcano plot showing the log2 fold change in expression between UPF1-overexpressing/UPF1-depleted and control-transfected cells. The horizontal line indicates the significance threshold (*p* < 0.05; *n* = 2). The black dots represent endogenous NMD target transcripts. (**B**) Expression levels of common transcripts that were upregulated and downregulated by UPF1 overexpression and UPF1 depletion, respectively, in Huh7 cells. (**C**) Validation of the mRNA expression level of DUSP1 in liver hepatocellular carcinoma (LIHC) tissues and normal liver tissues using the GEPIA database. (**D**) Huh7 cells transfected with MYC-UPF1 or MYC were treated with 100 μg/mL DRB for 2 h. The level of DUSP1 mRNA was measured at the indicated time point. The relative level of mRNA was normalized to that of GAPDH mRNA. (**E**) The level of DUSP1 mRNA in UPF1 variant-expressing Huh7 cells was measured by RT-qPCR. The level of mRNA was normalized to that of GAPDH mRNA. (**F**) Same as (**E**) except that the indicated proteins were evaluated by WB. (**G**) Immunofluorescence analysis was performed using Huh7 cells transfected with UPF1 variants (200× magnification). Immunofluorescence microscopy showed DUSP1, p27 and phopho-p53(S15) (green) and nuclei (blue). (**H**) The proposed model is shown. * *p* ≤ 0.05; ** *p* ≤ 0.01; *** *p* ≤ 0.001; ns, not significant.

## Data Availability

Raw RNA-seq data have been deposited in the NCBI Gene Expression Omnibus (GEO; https://www.ncbi.nlm.nih.gov/geo/, accessed on 13 October 2021) under accession number GSE185655.

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
