# Peer review of "UPF1 Inhibits Hepatocellular Carcinoma Growth through DUSP1/p53 Signal Pathway"

_biomedicines, 2022, doi:10.3390/biomedicines10040793_

Round 1

Reviewer 1 Report

The manuscript was improved and now can be accepted for publication

Reviewer 2 Report

The authors have convincingly addressed the concerns raised by the reviewer.

This manuscript is a resubmission of an earlier submission. The following is a list of the peer review reports and author responses from that submission.

Round 1

Reviewer 1 Report

In the present manuscript, Lee et al. investigated the role of UPF1 in human hepatocellular carcinoma (HCC). The authors found that  UPF1 overexpression reduces hepatocarcinogenesis in vitro and in vivo. Furthermore, the dual-specificity phosphatase 1 (DUSP1) tumor suppressor was identified as UPF1 specific target via posttranscriptional regulation at the molecular levels. Thus, UPF1 plays a tumor-suppressive role in hepatocarcinogenesis, at least partly by inducing DUSP1.

The study by Lee et al. is interesting, novel, and might provide important insights into the molecular mechanisms responsible for hepatocarcinogenesis. However, the investigation was not conducted in-depth, and some critical issues limit the importance of the present work. Specifically:

  1. The study was conducted only in one HCC cell line (HuH7). The experiments should have been conducted in additional HCC cell lines to rule out the possibility that the observed data are cell line-dependent.
  2. The authors studied the effects of UPF1 on the p53 pathway in the HuH7 cell line. However, the results should be considered irrelevant as HuH7 cells carry a p53 mutation that hampers p53 transcriptional activity. Thus, the conclusion that UPF1 acts by activating p53 cannot be drawn. HCC or hepatoblastoma cell lines with wild-type p53 should have been used instead.
  3. The role of UPF1 in terms of apoptosis should have been investigated.
  4. The authors claim that UPF1 is downregulated in HCC; however, the upregulation of UPF1 in HCC is reported by the Tumor Cancer Genome Atlas (http://ualcan.path.uab.edu/cgi-bin/TCGAExResultNew2.pl?genenam=UPF1&ctype=LIHC). How do the authors explain this discrepancy?

Reviewer 2 Report

In this manuscript named” UPF1 inhibits hepatocellular carcinoma growth through

DUSP1/p53 signal pathway” (1500672), the author is trying to explore the potential function in liver tumor (HCC) formation. UPF1 plays some functions in hepatocellular carcinoma (HCC) is well-known and many manuscript (above 8 papers) had shown it, largely decreasing the novelty of current manuscript. Many results are duplicated to previous papers, bot cell line is different (one cell line Huh 7 in manuscript). Also, in current manuscript, poor figure preparing, data un consistence and unclear statement in content decline the value of current manuscript. Further, according to the title, the author claimed UPF1 inhibited hepatocellular carcinoma growth through DUSP1/p53 signal pathway, but not detail approaches or validation is used to monitor the relevant effect instead of one immune blotting results. Although four Figures were showed in manuscript, to many un consistence points and conflict need to clarify from Figure1 a panel to 4g. Based on above opinion, I did not suggest this manuscript for publication in Biomedicines Journal.   

Reviewer 3 Report

Suman Lee and co-authors present a quality and well-written experiments manuscript that describes inhibition of hepatocellular carcinoma growth by UPF1 through DUSP1/p53 signaling pathway.

Authors aimed to clarify the function of UPF1 in hepatocellular carcinoma proliferation as a key nonsense-mediated mRNA (NMD) decay factor. For that they evaluated the effects of UPF1 overexpression on NMD and Huh7 cell growth.

Authors observed that overexpression of UPF1 did not change the levels of exogenous and endogenous NMD targets but did retard Huh7 cell growth. Moreover, overexpression of UPF1 deletion variants with loss of the key domains for proper NMD also reduced Huh7 tumorigenesis.

Authors confirmed all in vitro results in a xenograft model, indicating that UPF1 variant-expressing tumor tissues exhibited loss of tumorigenesis. Transcriptome analysis of cells with UPF1 overexpression or depletion identified the potential UPF1-targeted transcript DUSP1, which is posttranscriptionally regulated by UPF1.

Finally, authors conclude that the growth of liver cancer can be slowed by UPF1 variant overexpression, not through UPF1-mediated NMD.

===========================

Other comments:

1) Please check for typos throughout the manuscript.

2) The conclusion or concluding paragraph seems to be missing. Please add and briefly mention future prospectives in the field that are relevant to your study.

3) Regarding activation of p53 pathway - authors are kindly encouraged to cite this article that describes various aspects of developing novel therapeutic p53 activators for the treatment of carcinoma.
DOI: 10.22099/mbrc.2019.34179.1419

Overall, the manuscript is valuable for the scientific community and should be accepted for publication after edits are made.